# The Role of *IL-9* Polymorphisms and Serum IL-9 Levels in Carcinogenesis and Survival Rate for Laryngeal Squamous Cell Carcinoma

**DOI:** 10.3390/cells10030601

**Published:** 2021-03-09

**Authors:** Agne Pasvenskaite, Rasa Liutkeviciene, Greta Gedvilaite, Alvita Vilkeviciute, Vykintas Liutkevicius, Virgilijus Uloza

**Affiliations:** 1Department of Otorhinolaryngology, Lithuanian University of Health Sciences (LUHS), LT-50161 Kaunas, Lithuania; vykintas.liutkevicius@lsmuni.lt (V.L.); virgilijus.ulozas@lsmuni.lt (V.U.); 2Neuroscience Institute, Lithuanian University of Health Sciences (LUHS), LT-50161 Kaunas, Lithuania; rasa.liutkeviciene@lsmuni.lt (R.L.); greta.gedvilaite@lsmuni.lt (G.G.); alvita.vilkeviciute@lsmuni.lt (A.V.)

**Keywords:** laryngeal squamous cell carcinoma, *IL-9*, rs1859430, rs2069870, rs11741137, rs2069885, rs2069884, serum levels, survival rate

## Abstract

Recent studies have described the dichotomous function of IL-9 in various cancer diseases. However, its function has still not been analysed in laryngeal squamous cell carcinoma (LSCC). In the present study, we evaluated five single nucleotide polymorphisms (SNPs) of *IL-9* (rs1859430, rs2069870, rs11741137, rs2069885, and rs2069884) and determined their associations with the patients’ five-year survival rate. Additionally, we analysed serum IL-9 levels using an enzyme-linked immunosorbent assay. Three hundred LSCC patients and 533 control subjects were included in this study. A significant association between the patients’ survival rate and distribution of *IL-9* rs1859430 variants was revealed: patients carrying AA genotype had a higher risk of dying (*p* = 0.005). Haplotypes A-G-C-G-G of *IL-9* (rs1859430, rs2069870, rs11741137, rs2069885, and rs2069884) were associated with 47% lower odds of LSCC occurrence (*p* = 0.035). Serum IL-9 levels were found detectable in three control group subjects (8.99 ± 12.03 pg/mL). In summary, these findings indicate that the genotypic distribution of *IL-9* rs1859430 negatively influences the five-year survival rate of LSCC patients. The haplotypes A-G-C-G-G of *IL-9* (rs1859430, rs2069870, rs11741137, rs2069885, and rs2069884) are associated with the lower odds of LSCC development.

## 1. Introduction

Laryngeal squamous cell carcinoma (LSCC) continues to be the most frequent malignancy of head and neck squamous cell carcinoma (HNSCC) and is the most commonly diagnosed oncological disease in otorhinolaryngology [1]. Furthermore, LSCC is second only to pulmonary cancer in terms of mortality in all respiratory tract cancers [2]. According to Global Cancer Observatory data, 184,615 new cases of LSCC worldwide and 99,840 cases of death were recorded in 2020 [3].

Although sophisticated diagnostic tools (video laryngostroboscopy, flexible endoscopy, or/and contact endoscopy) are utilised to diagnose LSCC, unfortunately, approximately 75% of these patients are diagnosed in advanced stages (stages III and IV) [4]. Delays in diagnosing LSCC could be explained by seemingly irrelevant symptoms such as hoarseness at the beginning of the disease, as only severe symptoms of later stages (laryngeal stenosis with dyspnoea and stridor, dysphagia, otalgia, and haemoptysis) lead these patients to doctor’s consultation [5]. Additionally, the lack of screening programs (endoscopic laryngeal examination performed by an otorhinolaryngologist) results in a delayed diagnosis of LSCC [6]. Hence, a better understanding of LSCC cellular heterogeneity could contribute to improving early diagnostics and survival rates of LSCC patients [7]. To find new blood- or tissue-based biomarkers and to assess their association with the morphological and clinical manifestations of LSCC, recent research has focused on the role of genetic susceptibility factors as single nucleotide polymorphisms in carcinogenesis [8,9,10].

Interleukin-9 (IL-9) is a pleiotropic cytokine whose gene is located on the long arm of chromosome 5 [11]. Specific secretion of IL-9 is observed not only in T helper type 2 (Th2) cells but also in activated Th9 cells, Th17 cells, regulatory T cells, and mast cells [12]. Moreover, IL-9 might be produced by eosinophils, neutrophils, innate lymphoid cells, natural killer T cells, and dendritic cells [13,14,15,16,17]. The IL-9 receptor consists of two subunits: the alpha-chain (IL-9Rα), which is bound to Janus Kinase 1 (JAK1), and the common gamma chain (γ_c_), which is bound to JAK3 [18]. When IL-9 binds to its receptor, activation of JAK1 and JAK3 kinases is induced, and as a result, signal transducer and activator of transcription 1 (STAT-1), STAT-3, STAT-5 pathways are triggered [19]. Gene expression of IL-9 is controlled by phosphorylated STAT molecules in the nucleus, where they bind to regulatory sequences and provide pleiotropic biological functions of IL-9 [11,20].

It has been clarified that IL-9 plays an important role in different inflammation processes and autoimmune diseases [21]. Interestingly, recent research has suggested that IL-9 can have both a protumorigenic and antitumorigenic role in the pathogenesis of neoplasia [11,21,22,23]. This dichotomous function of IL-9 has been described in various cancer diseases. The protumorigenic activation—including proliferation and growth of cancer cells through the JAK/STAT pathway—has been suggested in carcinogenesis of different immune system organs: hepatocellular cancer, lung, breast, colorectal, and thyroid cancers [24,25,26,27,28]. Depending on a cell line and tumour microenvironment, IL-9 demonstrates anticancer behaviour: IL-9 inhibits cell growth and initiates apoptosis in melanoma tissues; the release of tumour necrosis factor-alpha (TNFα), IL-2 and interferon-gamma (INFγ) suggests antitumorigenic influence on lung cancer; by regulating T-cell function and killing tumour cells, IL-9 plays an antitumorigenic role in colon cancer [29,30,31]. However, it is necessary to identify the cellular origin of IL-9, as it is still not determined which cell types are susceptible to its production in most cancer diseases [20].

To the best of our knowledge, the relationship between *IL-9* rs1859430, rs2069870, rs11741137, rs2069885, and rs2069884 single nucleotide polymorphisms (SNPs), serum IL-9 levels, and LSCC carcinogenesis has not been analysed in the scientific literature.

## 2. Materials and Methods

The present case–control study was conducted at the Department of Otorhinolaryngology, Lithuanian University of Health Sciences (LUHS), Kaunas, Lithuania, and at the Laboratory of Ophthalmology, Neuroscience Institute, LUHS, Kaunas, Lithuania. 

### 2.1. Ethics Statement

The study protocol was confirmed by Kaunas Regional Ethics Committee for Biomedical Research, LUHS (authorisation number BE-2-37). All study concerning procedures were accomplished following the Declaration of Helsinki. All participants were informed about the structure and objectives of the present study before its launch. An Informed Consent Form was obtained from all subjects involved in the study.

### 2.2. Study Population

The present study enrolled 833 subjects: 300 patients with LSCC, and 533 healthy controls as a reference group. The characteristics of study subjects are given in Table 1. Data on age, sex, smoking habits, and alcohol consumption were compared between the LSCC and control groups. The control group was adjusted by sex and age to the LSCC group (*p* = 0.813; *p* = 0.054, respectively).

### 2.3. Selection of Study Population

The study population was divided into two groups: the LSCC group and healthy controls.

LSCC group. A comprehensive otorhinolaryngological examination including flexible endoscopy and/or video laryngostroboscopy and neck palpation was carried out for all patients with suspected LSCC at the Outpatient Office of the Department of Otorhinolaryngology, LUHS. Collection of peripheral venous blood samples was performed from the catheter inserted to induce general anaesthesia. All the patients underwent direct microlaryngoscopy with biopsy. Histological diagnosis of LSCC was confirmed at the Department of Pathology, LUHS. Other important clinical data were obtained by reviews of patients’ case records and personal interviews. Data about the harmful habits were collected from the unified questionnaire used in the Health Behavior among Lithuanian Adult Population, 2012 project (questions from numbers 50 to 62 were selected for the study) [32]. To determine the final diagnosis with staging, laryngeal and neck CT scan or/and MRI were performed. The staging of LSCC was done following the Guidelines for Head and Neck Cancers Classification, Version 2.2020 accepted by National Comprehensive Cancer Network (NCCN) [33].

Patients diagnosed with another type and localisation of cancer, acute or chronic infectious diseases, individuals using psychomotor suppressants and antiepileptic drugs, pregnant or breastfeeding woman, persons younger than 18 years old (according to Convention on the Rights of the Child (Lithuania acceded in 1992), every human being below the age of 18 means a child if his or her adulthood has not been recognised before by law) were excluded from this study [34].

Healthy controls. Patients who presented at the otorhinolaryngologist’s consultation at the Out-patient Office at the Department of Otorhinolaryngology, LUHS, and were selected for surgical treatment (tympanoplasty, ossiculoplasty, tympanostomy, nasal bones reposition, septoplasty, rhinoseptoplasty, uvulopalatopharyngoplasty, or radiofrequency thermoablation) were enrolled into the present study. Peripheral venous blood samples were collected from the same catheter inserted to induce general anaesthesia. Additionally, patients who presented at the family doctor’s consultation for a general check-up and had a complete blood count test were enrolled in this study. Patients with the diagnosed oncologic disease, acute or chronic infectious diseases, individuals using psychomotor suppressants and antiepileptic drugs, pregnant or breastfeeding women, and persons younger than 18 years old were excluded from this study.

### 2.4. SNP Selection

Molecular characteristics and mechanisms of LSCC remain unrevealed. However, it has been clarified that environmental carcinogens can induce damage of DNA and cause genomic irregularity [35,36,37,38,39,40]. Knowing that most LSCC patients are heavy smokers and alcohol consumers, studies analysing the molecular landscape and heterogeneity of LSCC, based on genetic alterations, could contribute to better understanding of the pathogenesis of this disease [7].

Recently, the impact of IL-9 on immune regulation function in the development of different types of tumours has been suggested [22]. In accordance with these findings, for the present study we selected different SNPs of *IL-9:* rs1859430, rs2069870, rs11741137, rs2069885, and rs2069884, to determine possible associations with LSCC development. The selected *IL-9* SNPs are frequently analysed in scientific literature.

*IL-9:* rs1859430, rs2069870, rs11741137, rs2069885, and rs2069884

Allergic and infectious diseases. Based on the evidence that IL-9 is produced by mast cells and Th2 cells during inflammatory responses, it has been suggested that IL-9 plays a significant role in inflammatory reactions of the airway [13]. For this reason, *IL-9* polymorphisms are mostly investigated in allergic and infectious respiratory tract diseases. Moreover, *IL-9* polymorphisms are also linked to sex-specific differences: *IL-9 rs2069885* demonstrates increased susceptibility to severe respiratory syncytial virus infection in girls [41]; specific sex associations of *IL-9 rs2069885* were also demonstrated with asthma in males [42]. These findings suggest that differences in immune-related genes, due to SNPs, have a nonidentical effect on the severity of disease, and indicate the importance of heterogeneity according to sex and pleiotropy [41,42]. Significant manifestations of *IL-9* rs1859430, rs11741137, rs2069885 dominant genotypes in asthma subjects demonstrated that these patients were more likely to have a severe asthma exacerbation caused by elevated dust mite exposition [43]. Additionally, it has been suggested that rs731476 T-/rs2069885 G-genotype combination in the *IL9R*/*IL9* encoding genes indicates a significantly higher risk of allergic rhinitis development in women [44].

Other diseases. Identification of *IL-9* polymorphisms was accomplished in autoimmune diseases suggesting that rs1859430 polymorphisms could be the cogenetic risk factor for Graves’ disease and Graves’ ophthalmopathy [45]. Schürks et al. have suggested a possible association of *IL-9* rs2069885 with an increased risk for active migraine without aura [46].

Cancer. Recently, the importance of IL-9 and IL-9 producing cells in tumour immunity has captivated attention. A significant association of *IL-9* rs1859430 was demonstrated both with pituitary adenoma development and with recurrence of the disease; however, no statistically significant differences between the rs206987 variant distribution were observed [47]. Given the dichotomous role of IL-9 in carcinogenesis, more studies analysing particular oncological diseases are required [20].

With the reference to a convincing role of IL-9 in respiratory tract diseases and existing sex-specific associations of *IL-9* SNPs, these arguments lead us to the hypothesis that *IL-9* SNPs might play a significant role in carcinogenesis when the oncologic process is located in the airway and be predominant in males, like in LSCC patients.

To the best of our knowledge, no comprehensive studies analysing the associations of *IL-9* SNPs rs1859430, rs2069870, rs11741137, rs2069885, and rs2069884 with LSCC development have been reported in the scientific literature.

### 2.5. DNA Extraction

DNA extraction was carried out at the Laboratory of Ophthalmology, Neuroscience Institute, LUHS. The DNA was extracted from peripheral venous blood samples (leucocytes) collected in 200 µL test-tubes utilising the silica-based membrane technology, using a genomic DNA extraction kit (GeneJET Genomic DNA Purification Kit, Thermo Fisher Scientific, Vilnius, Lithuania), based on manufacturer’s recommendations.

### 2.6. Genotyping

The analysis of *IL-9* gene polymorphisms rs1859430, rs2069870, rs11741137, rs2069885, and rs2069884 was carried out at the Laboratory of Ophthalmology, Neuroscience Institute, LUHS. The genotyping of *IL-9* (rs1859430, rs2069870, rs11741137, rs2069885, and rs2069884) was performed using the real-time polymerase chain reaction (PCR) method. Identification of all single-nucleotide polymorphisms was performed using TaqMan^®^ Genotyping assays (Thermo Fisher Scientific, Inc, Pleasanton, USA). The genotyping was performed using a “StepOnePlus” real-time PCR quantification system (Thermo Fisher Scientific, Singapore). Results on individual genotypes were obtained using the Allelic Discrimination program during the real-time PCR.

### 2.7. Serum IL-9 Levels Measurement

Serum IL-9 levels were evaluated in 20 control subjects and 20 LSCC patients. The assay was performed using an Invitrogen ELISA Kit (Cat. No. BMS2081) for human IL-9, standard curve sensibility range: 3.1–200 pg/mL, sensitivity 0.5 pg/mL, following the manufacturer’s instructions, and analysed with a Multiskan FC microplate photometer (Thermo Scientific, Waltham, MA) at 450 nm.

### 2.8. Quality Control of Genotyping

The repetitive analysis of 5% randomly chosen samples was performed for all five SNPs to confirm the same rate of genotypes from initial and repetitive genotyping.

### 2.9. Survival Rate

The data of the LSCC group about the mortality rate, including the survival period after diagnosis of LSCC, and the cause of death were collected from the Lithuanian State Register of Death Cases and Their Causes.

### 2.10. Statistical Analysis

Data on demographic characteristics of study participants were compared between control group subjects and the LSCC group using the Pearson Chi-square test and Student’s *t*-test and presented as absolute numbers with percentages in brackets. The frequencies of all selected *IL-9* SNPs genotypes and alleles are presented in percentages.

To compare the observed and expected frequencies of *IL-9* polymorphisms rs1859430, rs2069870, rs11741137, rs2069885, and rs2069884 in the control group, the analysis of Hardy–Weinberg using the Chi-square test was performed. The Chi-square test was used to compare distribution of *IL-9* rs1859430, rs2069870, rs11741137, rs2069885, and rs2069884 SNPs in the LSCC and control groups. Binomial logistic regression analysis with an adjusted odds ratio (OR) and its 95% confidence interval (95% CI) was utilised to evaluate the influence of *IL-9* rs1859430, rs2069870, rs11741137, rs2069885, and rs2069884 genotypes and alleles on LSCC development, and the risk prediction for LSCC patients with these polymorphisms. The binomial logistic regression analysis results are represented as genetic models: codominant, dominant, recessive, overdominant, and additive. The best genetic model selection was based on the Akaike Information Criterion (AIC); therefore, the best genetic models were those with the lowest AIC values.

Haplotype analysis was performed in the LSCC and control groups using online SNPStat software (https://www.snpstats.net/snpstats/, accessed 10 January 2021) [48]. Linkage disequilibrium (LD) was assessed by D’ and r^2^ measures. Associations between the haplotypes with frequencies of at least 1% and LSCC were calculated by logistic regression and presented as ORs, 95% CI and *p*-values.

LSCC patients’ survival analysis was accomplished using the Life-Table method. To compare survival rates in different subgroups, the Gehan’s criterion was utilised. To determine the impact of different variables on the risk of dying from LSCC, the univariate and multivariate *Cox* regression proportional hazard models were used. Different variables were evaluated analysing them one by one and as a whole of variables. Hazard ratios and their 95% confidence intervals were calculated.

Statistical analysis was performed using the SPSS/W 22.0 software (Statistical Package for the Social Sciences for Windows, Inc., Chicago, IL, USA). The findings were considered statistically significant when *p* < 0.05. Only statistically significant variables are presented in the tables.

## 3. Results

### 3.1. SNP Analysis

Hardy–Weinberg equilibrium (HWE) was assessed among control group subjects. The results demonstrated that *IL-9* genotypes rs1859430, rs11741137, rs2069885, and rs2069884 in the control group did not deviate from HWE (*p* > 0.05). However, we determined that *IL-9* rs2069870 was not in HWE (Table 2). According to Quality Control Procedures for Genome Wide Association Studies, SNPs out of HWE should not be removed from the analysis. Moreover, they should be flagged for further analysis after the association analysis is accomplished (e.g., MySQL Database, http://wb.mysql.com/, accessed 10 January 2021) [49].

In the present study, we carried out genotyping of *IL-9* rs1859430, rs2069870, rs11741137, rs2069885, and rs2069884 SNPs in 833 subjects (533 control group subjects and 300 LSCC patients) and analysed possible associations between the selected SNPs and LSCC development. Table 3 lists the frequencies of selected SNPs genotype profiles between the control group and LSCC patients. However, we did not find any statistically significant differences between the control group and LSCC patients (Table 3).

To evaluate the impact of *IL-9* polymorphisms rs1859430, rs2069870, rs11741137, rs2069885, and rs2069884 on LSCC development, a binomial logistic regression was applied. However, the results did not show a statistically significant impact of the selected SNPs on LSCC development (Appendix A).

Trying to better understand the heterogeneity and spreading of LSCC, we divided the LSCC group into the early stage (stages I and II) subgroup and advanced (stages III and IV) subgroup. Statistical analysis did not reveal any significant differences between the genotype distribution of control and LSCC groups (Appendix A).

In addition, we did not detect any statistically significant differences dividing the LSCC group into subgroups of patients with no metastasis and with metastasis to the neck lymph nodes (Appendix A).

### 3.2. Haplotype Association with the Predisposition to LSCC Occurrence

Strong linkage disequilibrium between studied *IL-9* SNPs: rs1859430, rs2069870, rs11741137, rs2069885, and rs2069884 was observed (Table 4).

While the haplotypes analysis identified many of their sets, differences in the haplotype frequencies between the LSCC and control groups were observed (Table 5). The results of the frequencies of haplotypes among patients with LSCC and controls showed that haplotypes A-G-C-G-G of *IL-9* rs1859430, rs2069870, rs11741137, rs2069885, and rs2069884 are associated with 47% decreased odds of LSCC occurrence (OR = 0.53; 95% CI:0.30–0.95, *p* = 0.035) (Table 5).

### 3.3. Serum Concentrations of IL-9 in the Control and LSCC Groups

Serum IL-9 levels were measured in 20 control subjects and 20 LSCC patients. The results were obtained from all 40 subjects. However, significant IL-9 levels, according to measuring range 3.1–200 pg/mL and sensitivity 0.5 pg/mL, were identified only in three control group subjects with the mean of 8.99 ± 12.03 pg/mL.

### 3.4. Survival Analysis

The five-year overall survival (OS) rate of selected 300 LSCC patients including all causes of death was 65% (Figure 1). The percentage of LSCC group surviving exactly from the LSCC diagnosis date—LSCC-specific survival—to a period of five years was 75% (patients who died from causes unrelated to LSCC were not included in this measurement).

Analysing LSCC patients’ five-year survival rate and the genotype distribution of *IL-9* SNPs rs1859430, rs2069870, rs11741137, rs2069885, and rs2069884 we clarified that subjects carrying AA genotype at rs1859430 had a statistically significantly poorer five-year survival rate (five-year LSCC-specific survival 45%) than those carrying AG and GG genotypes (five-year LSCC-specific survival 69% and 65%, respectively; *p* = 0.026; Figure 2).

Table 6 represents the impact of genotype distribution of the selected SNPs on the survival rate.

According to the results from the univariate *Cox* proportional hazards model, different characteristics were analysed one by one, without excluding the influence of other variables. Tumour size (T) and metastasis to the neck lymph nodes (N) (according to TNM classification [33]), tumour differentiation grade (G), stage, smoking experience, and distribution of *IL-9* rs1859430 genotypes were effective variables in the LSCC-specific survival. It was clarified that patients with T2, T3, and T4 had 6.879, 6.575, and 7.644 times increased risk of dying, respectively, compared to patients with T1 (HR_T2 vs. T1_ = 6.879, 95% CI:2.747–17.228, *p* = 0.000; HR_T3 vs. T1_ = 6.575, 95% CI:2.592–16.681, *p* = 0.000; HR_T4 vs. T1_ = 7.644, 95% CI:3.134–18.644, *p* = 0.000). The risk of death of patients who had metastasis to the neck lymph nodes (N ≥ 1) was 2.969 times higher than of those without metastasis to the neck lymph nodes (N = 0) (HR_N ≥ 1 vs. N = 0_=2.969, 95% CI:1.808–4.877, *p* = 0.000). According to the tumour differentiation grade (G), it was clarified that patients with G3 have statistically significantly higher risk of death compared to patients with G1 and G2 (HR_G3 vs. G1_ = 2.611, 95% CI:0.181–0.812, *p* = 0.023; HR_G3 vs. G2_ = 2.392, 95% CI:0.213–0.818, *p* = 0.011). According to the stage, the risk of dying of patients with stages II, III, and IV was higher than of those with stage I tumour (HR_II vs. I_ = 6.569, 95% CI:2.623–16.454, *p* = 0.000; HR_III vs. I_ = 6.423, 95% CI:22.491–16.559, *p* = 0.000; HR_IV vs. I_ = 7.169, 95% CI:2.959–17.371, *p* = 0.000). Patients who had ≥25 years of smoking experience had 2.372 times increased risk of death compared to those who were smoking for less than 25 years (HR_25≥ years vs. <25years_ = 22.372, 95% CI:0.915–6.147, *p* = 0.046). However, alcohol consumption increased the risk of death statistically insignificantly (HR_Users vs. Non-drinkers_ = 1.670, 95% CI:0.644–4.328, *p* = 0.291). Moreover, it was found that patients carrying AA genotype at *IL-9* rs1859430 had 2.503 times increased risk of death compared to those carrying AG and GG genotypes (HR_AA vs. AG and GG_ = 2.503, 95% CI:1.194–5.246, *p* = 0.015) (Table 7).

To analyse all of the variables, a multivariable *Cox* proportional hazards model was utilised. Variables that were analysed one by one previously and had a significant *p*-value were involved. To obtain an optimal model, the Backward Stepwise selection method was applied. Referring to the results after the Backward Stepwise selection method (the final multivariable *Cox* proportional hazards model), only three significant predictors remained: tumour size (T), metastasis to the neck lymph nodes, and *IL-9* rs1859430 AA genotype (Table 8). Moreover, although in univariable analysis (Table 7) inspecting different variables one by one, i.e., tumour differentiation grade (G) (G3 vs. G1 and G2), stage (I vs. II, III, and IV) and smoking experience were significant for patients’ survival rate, these characteristics were not significant in the multivariable analysis.

In multivariable analysis, it was identified that patients with T2, T3, and T4 had an increased risk of dying, compared to patients with T1 (HR_T2 vs. T1_ = 6.687, 95% CI:2.645–16.904, *p* = 0.000; HR_T3 vs. T1_ = 5.941, 95% CI:2.313–15.259, *p* = 0.000; HR_T4 vs. T1_ = 6.466, 95% CI:2.563–16.313, *p* = 0.000). Moreover, the risk of death in patients who had metastasis to the neck lymph nodes (N ≥ 1) was higher than in those without metastasis to the neck lymph nodes (N = 0) (HR_N ≥ 1 vs. N = 0_ = 1.850, 95% CI:1.099–3.113, *p* = 0.021). Furthermore, it was clarified that patients carrying AA genotype at *IL-9* rs1859430 have a 2.964 times higher risk of death compared to those carrying AG and GG genotypes (HR_AA vs. AG and GG_ = 2.964, 95% CI:1.397–6.285, *p* = 0.005) (Table 8).

## 4. Discussion

In this study, five *IL-9* SNPs: rs1859430, rs2069870, rs11741137, rs2069885, and rs2069884 were selected to analyse their possible role in the development of LSCC and to identify associations of genetic distribution of the selected SNPs with patients’ survival rate. We determined that patients carrying AA genotype at *IL-9* rs1859430 have a 2.503 times higher risk of death (*p* = 0.015) than those carrying AG and GG genotypes. Moreover, in our LSCC cohort, *IL-9* rs1859430 AA genotype remained in the optimal multivariable *Cox* proportional hazards model demonstrating a significant role in the prediction of LSCC survival (*p* = 0.005). Furthermore, haplotype analysis of the selected SNPs revealed that haplotypes A-G-C-G-G of *IL-9* (rs1859430, rs2069870, rs11741137, rs2069885, and rs2069884) are associated with 47 % lower odds of LSCC occurrence (*p* = 0.035). However, we did not detect any significant differences between genotypic distributions of *IL-9* rs1859430, rs2069870, rs11741137, rs2069885, and rs2069884 SNPs in the LSCC and control groups, as we expected selecting these polymorphisms based on their significant role in respiratory tract diseases and sex-specific associations [12,42].

The present study also confirmed the following significant characteristics for LSCC patients’ survival rate: tumour size (T), metastasis to the lymph nodes (N), tumour differentiation grade (G), and stage of LSCC. We determined that patients with T1 (*p* = 0.000), no metastasis to the neck lymph nodes (*p* = 0.000) and G1-2 (*p* = 0.023 and *p* = 0.011) had statistically significantly lower risk of death. Principally, these results are in concordance with data in the literature. However, the TNM classification system of the anatomical extent of the tumour does not take into account neither the biological aggressiveness of the particular tumour nor the immunological response of the host and is not oriented to personalised treatment [50].

Moreover, whereas oncologists are concentrated on optimal treatment options, patients are interested in their prognosis. For this reason, it is important to predict the probable outcome for an individual LSCC patient. Environmental factors as smoking and alcohol consumption might be these predictors. Berthiller et al. have suggested that low frequency of cigarette smoking (0–3 cigarettes per day) is related to a 50% increased risk of HNSCC development [51]. In the present study, we confirmed that patients who had smoked for longer than 25 years had a poor prognosis of LSCC and a 2.372 times higher risk of dying than those who had smoked for less than 25 years (*p* = 0.046). However, we did not find any significant differences between alcohol consumers and non-drinkers when analysing survival rate differences (*p* = 0.291). Bearing this in mind, the molecular biological markers might be better predictive factors for particular biological behaviour of LSCC and could suggest both prognosis and treatment options for an individual patient [50].

To analyse alterations in inflammatory pathways related to carcinogenesis, serum levels of various cytokines are investigated. An increased expression level of IL-9 in the serum has been observed as a negative prognostic factor in patients with Hodgkin’s lymphoma and extranodal NK/T-cell lymphoma [52,53]. Bussu et al. have analysed serum IL-9 levels in 31 HNSCC patients (16 patients of oral cavity SCC and 15 LSCC patients) and six healthy controls using multiplex biometric ELISA and have found that there were no statistically significant differences between selected groups (33.54 ± 18.85 vs. 46.59 ± 25.72 pg/mL, respectively) [54]. Several studies have found that the expression of IL–9 was higher in peripheral blood of patients with LSCC and small cell lung cancer than that of normal controls, and the expression levels of IL–9 in patients with stages III and IV were higher than those in patients with stages I and II, suggesting that IL–9 expression may correlate with tumour stages [55,56]. However, some studies have indicated that serum cytokine levels cannot be used as a potential biomarker in LSCC [57]. In the present study, we performed serum IL-9 concentration analysis in 20 control subjects and 20 LSCC patients. However, due to the assay range (3.1–200 pg/mL) and sensitivity (0.5 pg/mL), serum IL-9 levels could be measured only in three control subjects (8.99 ± 12.03 pg/mL.). Other studies have proved that serum IL-9 levels can be detected in peripheral blood. Thus, our results could be explained by insufficient number of subjects involved in the analysis or by the chosen method of serum IL-9 detection [54,55].

Despite improvements of LSCC treatment over past decades (robotic and laser technologies), the mortality rate of these patients remains high. In 2020, 99,840 cases of death caused by LSCC were registered, which is 5000 patients more than in 2018 (94,771) [3,58]. In this study, the LSCC-specific five-year survival rate was found to be 75%. It corresponds to International Head and Neck Cancer Epidemiology (INHANCE) consortium data, where the specific five-year survival rate for LSCC patients is 72.3% and is the highest reported overall survival rate across different anatomical sites of head and neck cancer [59].

According to the INHANCE consortium data, LSCC patients with less than or equal to high school educational level had worse overall survival than those who received more than high school education (HR = 2.54, 95%, CI:1.01–6.38) [59]. Not improving HNSCC survival rate might be also due to the highest incidence of suicide in all oncology populations [60]. Unknown prognosis of diagnosed oncological disease cause anxiety in the patients, and 40–57% of head and neck cancer patients develop the major depressive disorder at the time of diagnosis or during treatment [61]. For this reason, a huge interest is focused on the molecular landscape of LSCC, willing to diagnose this disease earlier and personalise the treatment. Therefore, according to the literature data, targeting IL-9 and the IL-9 pathway might be a potential strategy and a great promise for treating and fighting cancer [22].

The strength of this study was the involvement of a large study population (833 subjects in total), careful selection of investigated groups (both groups were adjusted for age and sex), the involvement of environmental risk factors (smoking and alcohol consumption), and collection of pure LSCC cohort. Most genetic studies presented in the scientific literature provide data united under the umbrella of the HNSCC term, including malignant tumours of different localisations (oral, pharyngeal, nasopharyngeal, hypopharyngeal, laryngeal regions, etc.) and ignoring that these malignancies have different aetiology, biological and clinical behaviour, and distinct genomic profiles [62]. Furthermore, we believe that pooling different cancer types together may mask possible significant associations of selected biomarkers with individual cancer types.

To the best of our knowledge, this is the first report that associates the role of *IL-9* rs1859430, rs2069870, rs11741137, rs2069885, and rs2069884 SNPs and development of LSCC in wide-ranging, pure and homogenous LSCC patients’ cohort and age- and sex-matched control subjects. This uniqueness enabled us to perform a precise analysis of associations between selected SNPs and LSCC development and of survival rate with a particular tumour in a specific head and neck anatomical region. According to already confirmed and analysed LSCC behaviour, LSCC is presented as less aggressive compared to other HNSCC, assuming rather low metastatic rate and local spreading [63]. Therefore, the results of the present study showing the absence of differences in *IL-9* rs1859430, rs2069870, rs11741137, rs2069885, and rs2069884 variants distribution between LSCC tumour patients and sex-matched control subjects are apprehensible. However, a statistically significant association of AA genotype at *IL-9* rs1859430 with poorer LSCC-specific five-year survival rate, as well as the association of haplotypes A-G-C-G-G of *IL-9* (rs1859430, rs2069870, rs11741137, rs2069885, and rs2069884) with decreased odds of LSCC occurrence, suggest the importance of these SNPs and haplotypes in the development of LSCC.

## 5. Conclusions

Results of the present study demonstrate a significant association between haplotypes A-G-C-G-G of *IL-9* rs1859430, rs2069870, rs11741137, rs2069885, rs2069884 and the decreased risk for LSCC development. Additionally, it indicates that AA genotype at *IL-9* rs1859430 in LSCC patients is associated with a poorer five-year survival rate. However, the role of IL-9 in cancer including LSCC is not completely understood yet. Further research is required before it can be used as a therapeutic target.

## Figures and Tables

**Figure 1 cells-10-00601-f001:**
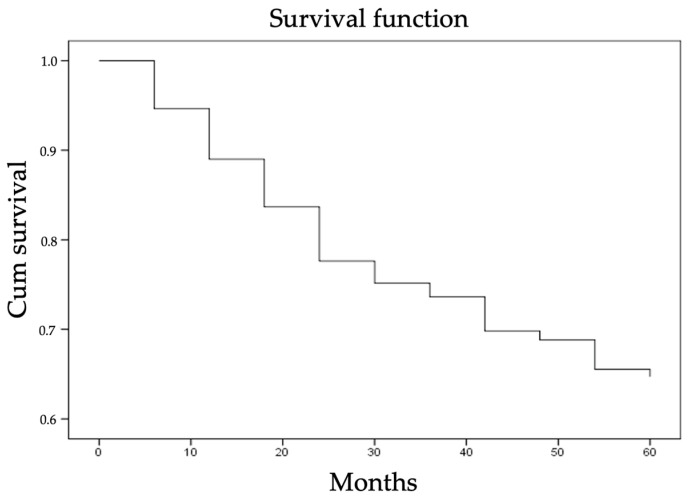
The five-year overall survival rate of the patients included in the LSCC group.

**Figure 2 cells-10-00601-f002:**
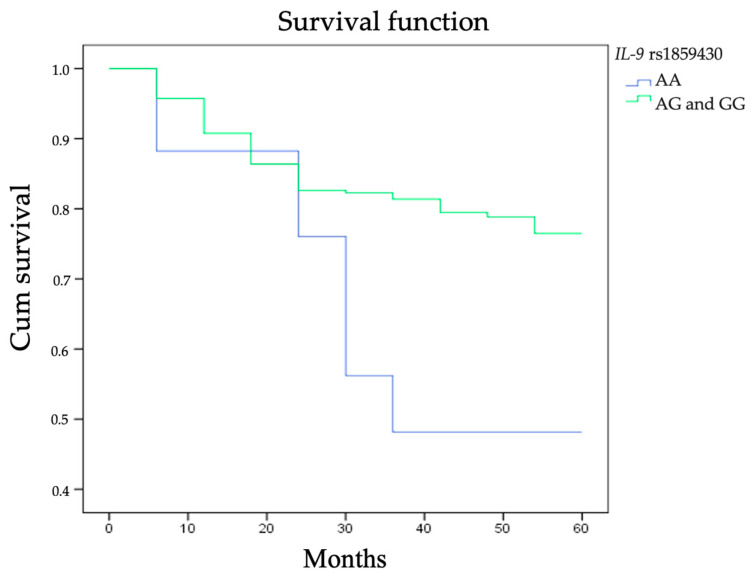
Five-year survival rate according to the distribution of *IL-9* rs1859430 genotypes.

**Table 1 cells-10-00601-t001:** Demographic characteristics of the study.

Characteristic	Group		*p*-Value *^7^*
	LSCC ^1^ *n* = 300	Control Group *n* = 533	
Male, *n* (%)	287 (95.7)	508 (95.3)	0.813 *
Female, *n* (%)	13 (4.3)	25 (4.7)	
Age years; mean (SD) ^2^	62.4 (9.6)	63.7 (9.6)	0.054 **
Smoking, *n*			<0.001
Yes	132 (44.0)	36 (6.8)	
No	5 (1.7)	166 (31.1)	
Unknown	163 (54.3)	331 (62.1)	
Alcohol consumption ***, *n*			<0.001
Yes	107 (35.7)	108 (20.3)	
No	30 (10.0)	94 (17.6)	
Unknown	163 (54.3)	331 62.1)	
Stage, *n* (%)		-	-
I	109(36.3)		
II	60 (20.0)		
III	50 (16.7)		
IV	81 (27.0)		
T ^3^, *n* (%)		-	-
1	112 (37.3)		
2	58 (19.3)		
3	57 (19.1)		
4	73 (24.3)		
N ^4^, *n* (%)		-	-
0	242 (80.7)		
1	18 (6.0)		
2	39 (13.0)		
3	1 (0.3)		
M ^5^, *n* (%)			
0	297 (99.0)		
1	2 (0.7)		
2	1 (0.3)		
G ^6^, *n* (%)		-	-
1	90 (30.0)		
2	183 (61.0)		
3	27 (9.0)		

^1^ LSCC: Laryngeal Squamous Cell Carcinoma; ^2^ SD: standard deviation; ^3^ T: tumour size; ^4^ N: metastasis to the neck lymph nodes; ^5^ M: metastasis; ^6^ G: tumour differentiation grade; ^7^
*p*-Value: significance level *p* < 0.05; * Pearson Chi-Square; ** Student’s *t*-test. *** Data about smoking and alcohol consumption were collected from 138 LSCC patients and 202 control group subjects.

**Table 2 cells-10-00601-t002:** Analysis of Hardy–Weinberg equilibrium in the control group.

SNP ^1^	Gene Location	Allele Frequencies		Genotype Distribution	*p*-Value ^2^
*IL-9* rs1859430		0.78 G	0.22 A	28/183/322	0.764
*IL-9* rs2069870	5q3	0.80 A	0.20 G	0/208/325	<0.001
*IL-9* rs11741137		0.83 C	0.17 T	12/157/364	0.301
*IL-9* rs2069885	1.1	0.83 G	0.17 A	1/155/367	0.248
*IL-9* rs2069884		0.83 G	0.17 T	1/155/367	0.248

^1^ SNP: single nucleotide polymorphism; ^2^
*p*-Value: significance level *p* < 0.05.

**Table 3 cells-10-00601-t003:** Frequencies of genotypes and alleles of *IL-9* rs1859430, rs2069870, rs11741137, rs2069885, and rs2069884 in the control and LSCC groups.

Polymorphism	Control Group *n* (%) (*n* = 533)	LSCC ^1^ *n* (%) (*n* = 300)	*p*-Value ^2^
*IL-*9 *rs1859430*			0.730
G/G	322 (60.4)	188 (62.7)	
G/A	183 (34.3)	95 (31.7)	
A/A	28 (5.3)	17 (5.7)	
Total	533 (100)	300 (100)	
Allele			
G	827 (77.6)	471 (78.5)	
A	239 (22.4)	129 (21.5)	
*IL-9* *rs2069870*			0.443
A/A	325 (61.0)	191 (63.7)	
A/G	208 (39.0)	109 (36.3)	
G/G	-	-	
Total	533 (100)	300 (100)	
Allele			
A	858 (80.5)	491 (81.8)	
G	208 (19.5)	109 (18.2)	
*IL-9* *rs11741137*			0.633
C/C	364 (68.3)	210 (70.0)	
C/T	157 (29.5)	81 (27.0)	
T/T	12 (2.3)	9 (3.0)	
Total	533 (100)	300 (100)	
Allele			
C	885 (83.0)	501 (83.5)	
T	181 (17.0)	99 (16.5)	
*IL-9* *rs2069885*			0.744
G/G	367 (68.9)	213 (71.0)	
G/A	155 (29.1)	78 (26.0)	
A/A	11 (2.1)	9 (3.0)	
Total	533 (100)	300 (100)	
Allele			
G	889 (83.4)	504 (84.0)	
A	177 (16.6)	96 (16.0)	
*IL-9* *rs2069884*			0.744
G/G	367 (68.9)	213 (71.0)	
G/T	155 (29.1)	78 (26.0)	
T/T	11 (2.1)	9 (3.0)	
Total	533 (100)	300 (100)	
Allele			
G	889 (83.4)	504 (84.0)	
T	177 (16.6)	96 (16.0)	

^1^ LSCC: laryngeal squamous cell carcinoma; ^2^
*p*-Value: significance level *p* < 0.05.

**Table 4 cells-10-00601-t004:** Linkage disequilibrium between *IL-9* rs1859430, rs2069870, rs11741137, rs2069885, and rs2069884 SNPs.

	*IL-9 rs1859430* (D’; r^2^) ^1,2^	*IL-9 rs2069870* (D’; r^2^) ^1,2^	*IL-9 rs11741137* (D’; r^2^) ^1,2^	*IL-9 rs2069885* (D’; r^2^) ^1,2^	*IL-9 rs2069884* (D’; r^2^) ^1,2^
*IL-9 rs1859430* (D’; r^2^) ^1,2^		0.9236; 0.7071	0.8768; 0.5477	0.8987; 0.5583	0.8987 0.5583
*IL-9 rs2069870* (D’; r^2^) ^1,2^			0.7937; 0.5417	0.8273; 0.5708	0.8273; 0.5708
*IL-9 rs11741137* (D’; r^2^) ^1,2^				0.9777; 0.9272	0.9777; 0.9272
*IL-9 rs2069885* (D’; r^2^) ^1,2^					0.9995; 0.999
IL-9 *rs2069884* (D’; r^2^) ^1,2^					

^1^ D’: linkage disequilibrium coefficient; ^2^ r^2^: squared correlation coefficient of the haplotype frequencies [r^2^ scale: 0.1].

**Table 5 cells-10-00601-t005:** Haplotype association with the predisposition to LSCC occurrence.

	*IL-9* *rs1859430*	*IL-9* *rs2069870*	*IL-9* *rs11741137*	*IL-9* *rs2069885*	*IL-9* *rs2069884*	Frequency	OR ^2^ (95 % CI ^3^)	*p*-Value ^4^
1	A	G	T	A	T	0.1332	1.01 (0.72–1.41)	0.95
2	A	G	C	G	G	0.0439	0.53 (0.30–0.95)	0.035
3	A	A	C	G	G	0.0237	1.26 (0.65–2.45)	0.49
4	A	A	T	A	T	0.017	0.78 (0.33–1.85)	0.57
5 (Rare) ^1^	*	*	*	*	*	0.0266	0.74 (0.37–1.49)	0.40

^1^ Rare: pooled rare haplotypes; ^2^ OR: odds ratio; ^3^ CI: confidence interval; ^4^
*p*-Value: significance level *p* < 0.05; *: all the haplotypes with low frequency were pooled and analysed as one haplotype group.

**Table 6 cells-10-00601-t006:** The genotype distribution of *IL-9:* rs1859430, rs2069870, rs11741137, rs2069885, and rs2069884, according to the 1-, 3-, and 5-year LSCC-specific survival rate.

Polymorphism	Genotype	*N* = 300 (%)	Survival Rate (%)			*p*-Value ^2,^*
			1-Year LSCC ^1^-Specific Survival (%)	3-Year LSCC ^1^-Specific Survival (%)	5-Year LSCC ^1^-Specific Survival (%)	
*IL-9* rs1859430	AA	17 (5.7)	88	45	45	0.188 0.026 **
	AG	95 (31.7)	81	74	69	
	GG	188 (62.6)	85	71	65	
*IL-9* rs2069870	AA	191 (63.7)	84	71	64	0.853
	AG	109 (63.7)	83	68	66	
*IL-9* rs1174113	CC	210 (70.0)	85	71	65	0.619
	CT	81 (27.0)	79	67	64	
	TT	9 (3.0)	100	76	76	
*IL-9* rs2069885	AA	9 (3.0)	100	76	76	0.640
	GA	78 (26.0)	79	67	64	
	GG	213 (71.0)	85	71	65	
*IL-9* rs2069884	GG	213 (71.0)	85	71	65	0.640
	GT	782 (26.0)	79	67	64	
	TT	9 (3.0)	100	76	76	

^1^ LSCC: laryngeal squamous cell carcinoma; ^2^
*p*-Value: significance level *p* < 0.05; * Gehan’s criterion; ** AA vs. AG and GG.

**Table 7 cells-10-00601-t007:** Association between study variables and LSCC patients’ mortality in univariate *Cox* proportional hazards model.

Variable		HR ^1^	Univariate 95% CI ^2^	*p*-Value ^6^
T ^3^	T2 vs. T1	6.879	2.747–17.228	0.000
	T3 vs. T1	6.575	2.592–16.681	0.000
	T4 vs. T1	7.644	3.134–18.644	0.000
N ^4^	N≥1 vs. N = 0	2.969	1.808–4.877	0.000
G ^5^	G3 vs. G1	0.383	0.181–0.812	0.023
	G3 vs. G2	0.418	0.213–0.818	0.011
Stage	II vs. I	6.569	2.623–16.454	0.000
	III vs. I	6.423	2.491–16.559	0.000
	IV vs. I	7.169	2.959–17.371	0.000
Smoking ≥25 years vs. <25 years		2.372	0.915–6.147	0.046
Alcohol Users vs. Non-drinkers		1.670	0.644–4.328	0.291
*IL-9* rs1859430 AA vs. AG and GG		2.503	1.194–5.246	0.015

^1^ HR: hazard ratio; ^2^ CI: confidence interval; ^3^ T: tumour size; ^4^ N: metastasis to the neck lymph nodes; ^5^ G: tumour differentiation grade; ^6^
*p*-Value: significance level *p* < 0.05.

**Table 8 cells-10-00601-t008:** Association between study variables and LSCC patients’ mortality in the final multivariable *Cox* proportional hazards model after applying the Backward Stepwise selection method.

Variable		HR ^1^	Multivariable 95% CI ^2^	*p*-Value ^5^
T ^3^	T2 vs. T1	6.687	2.645–16.904	0.000
	T3 vs. T1	5.941	2.313–15.259	0.000
	T4 vs. T1	6.466	2.563–16.313	0.000
N ^4^	N ≥ 1 vs. N = 0	1.850	1.099–3.113	0.021
*IL-9* rs1859430 AA vs. AG and GG		2.964	1.397–6.285	0.005

^1^ HR: hazard ratio; ^2^ CI: confidence interval; ^3^ T: tumour size; ^4^ N: metastasis to the neck lymph nodes; ^5^
*p*-Value: significance level *p* < 0.05.

## Data Availability

All data relevant to the study are included in the article or uploaded as Appendix A.

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
