# Peer review of "The Role of IL-9 Polymorphisms and Serum IL-9 Levels in Carcinogenesis and Survival Rate for Laryngeal Squamous Cell Carcinoma"

_cells, 2021, doi:10.3390/cells10030601_

Round 1
Reviewer 1 Report
In this study, Pasvenskaite et al analyzed the role of polymorphisms and serum levels of IL-9, as well as the expression of a regulatory microRNA - miRNA-891b in 300 patients with laryngeal squamous cell carcinoma (SCC) and 533 controls. They found a significant association between the patients’ survival rates and development of SCC, and distribution of some IL-9 single nucleotid polymorphisms (SNPs).
IL-9 is now believed to be an important cytokine not only in inflammatory diseases but in cancer as well. Its role in cancer is not well understood, and it seems to be complex, therefore new information regarding IL-9 in laryngeal SCC is valuable. The topic is thus attractive, and the results are potentially interesting. There are some issues that would need more attention which could contribute to the quality of the paper.
My main criticism is related to the length of the manuscript. It must be shortened and more focused on the topic of the research. There is no need to describe in details well known aspects of laryngeal SCC, such as epidemioloy, etiology and pathogenesis. Consistently, the second and third paragraphs of the Introduction can be shortened into a few sentences.
Similarly, the first paragraph of the Discussion can be shortened, as it describes (again) epidemiology. The second paragraph can also be shortened so that it does not repeat information regarding IL-9 which had already been described in the Introduction. It will be more attractive for the readers if the discussion starts immediately with the most important findings of this study, related to IL-9 SNPs.
Regulation of IL-9 expression by microRNAs is an important topic. However, as the authors analyzed only one microRNA - miRNA-891b and its expression in blood was not detected, I suggest to omit this result. More potential regulatory microRNAs should be analyzed to obtain relevant results, and it can be a topic of another study. The selection of regulatory microRNAs can be based on previous publications and miRNA and/or target gene data bases.
In the Conclusion, the authors state that “IL-9 SNPs could be …. potential prognostic factors in novel targeted and IL-9-based immunotherapy strategies ….”. Regarding the complex and dichotomous role of IL-9 in cancer, it is too early to speculate about IL-9 based immunotherapy. The contribution of this study is not relevant enough to support this conclusion. The conclusion must therefore be softened, clearly stating that the role of IL-9 in cancer including laryngeal SCC is not completely understood yet, and that much research is needed before it can be used as a therapeutic target.
Reviewer 2 Report
This manuscript presents the possible role of IL-9 polymorphisms to detect survival rate and haplotype association with the predisposition to LSCC occurrence. Moreover, IL-9 rs1859430 haplotype with AA contents showed the most inferior survival ratio with reliable statistic analysis. I firmly believe that this manuscript would provide new insight into the prediction of survival ratio and occurrence from the viewpoint of the IL-9 haplotype.
I like this manuscript very much because every basic research should give feedback to the clinical area, and this manuscript shows fascinating results for both researchers and clinicians.
I believe this manuscript would be suitable for publication after a small modification. Of course, authors should modify it following other reviewers.
Minor point:
- Page 9, line 350: Authors cited Table 10 in this line; however, there is no table 10 in this manuscript. Furthermore, I think it should be Table 5.
- Page 5, line 169: Please clarify the reason why patients younger than 18-year-old were excluded from this study. 18-year-old may be adult in your country; however, it is usually classified as juvenile generation, and there is no significant difference between 18 nor 20; therefore, some readers will be confused about the cut-off.
Reviewer 3 Report
Thank you for the opportunity to review this manuscript.
The authors showed that single-nucleotide polymorphisms (SNPs) of IL-9 rs1859430 negatively influence the five-year survival rate of patients with laryngeal squamous cell carcinoma (LSCC). Furthermore, the haplotypes A-G-C-G-G of IL-9 were found to be associated with a lower risk of LSCC development. These data indicate that IL-9 SNPs in serum may be a prognostic marker of LSCC.
I believe this manuscript is well written and includes several novel findings. This will be of interest to most LSCC researchers.
However, I have several points to improve this manuscript before publishing it in our journal.
- On page 9, line 350, the authors wrote “Table 10.” I think this is “Table 5” and not Table 10.
- On page 11, in the results section, it is difficult to understand which data in Tables 6 to 8 are described in the results section. It is better to indicate the table that is shown in the results section.
Round 2
Reviewer 1 Report
My suggestions have been addressed to adequately. I have no further comments.